# OpenReview forum: "Protein Autoregressive Modeling via Multiscale Structure Generation"
_ICLR.cc/2026/Conference — Submitted to ICLR 2026_

### Official Review · Reviewer_Zpd8 · 2025-10-23

**Soundness:** 4
**Presentation:** 3
**Contribution:** 3
**Rating:** 6
**Confidence:** 5

**Summary:**

This paper introduces Protein Autoregressive Modeling (PAR), a multi-scale, coarse-to-fine framework for protein backbone generation. Instead of predicting the next residue, PAR predicts the structure at the next scale of resolution, from a coarse layout to the final backbone.

This approach cleverly avoids two major limitations of standard AR models in this domain: (1) it avoids coordinate discretization by using a continuous, flow-based atomic decoder (Proteina), and (2) it avoids unidirectional bias by modeling global-to-local dependencies. The authors also identify and mitigate "exposure bias" using noisy context learning (NCL) and scheduled sampling (SS). The model demonstrates strong unconditional generation (competitive FPSD) and notable zero-shot generalization for tasks like prompt-based shape generation and motif scaffolding.

**Strengths:**

This is a high-quality, well-written, and clear paper that successfully applies a multi-scale autoregressive (AR) framework, similar to VAR in image generation, to the complex task of protein structure generation. The coarse-to-fine "sculpting" idea is interesting, and the paper's main contribution is in demonstrating this new path for autoregressive modeling in this domain.

**Weaknesses:**

However, the work is not without significant concerns. The methodology leans heavily on a large, pre-existing flow-based model (Proteina) as its core decoder, which makes it difficult to assess the true contribution of the AR-only components. Given the marginal improvements over the decoder baseline, the added complexity of the AR framework is not fully justified. The paper feels more like an incremental application of an existing algorithm to a new domain rather than a fundamental breakthrough.

**Questions:**

1. **Reliance on Flow-Based Decoder**: The method is presented as an "autoregressive model," but it heavily relies on a large, pre-trained flow-based model (Proteina) as its decoder. This blurs the lines of novelty. It feels less like a pure AR model and more like a complex, multi-scale conditioning scheme for a flow model.

2. **Questionable Benefit**: Given the relatively limited improvement over the Proteina baseline (Table 1), what is the practical benefit of introducing this complicated multi-scale AR scheme? The experiments do not convincingly demonstrate that this hybrid AR-flow approach is substantially better than the flow-based model it is built upon.

3. **Weaker Designability (sc-RMSD)**: While the model excels at global fold distribution (FPSD), its fine-grained designability (sc-RMSD) is a weakness. In Table 1, the PAR (400M) model's sc-RMSD of 1.28 is comparable to Proteina (1.09).

4. For evaluating diversity, the average pairwise TM-score (Table 1) is not very discriminative. A more robust metric, such as reporting the number of Foldseek clusters (as done in the Proteina paper), would be more convincing.

5. The interpretation of the attention maps (Fig. 6) concludes that each scale attends to the previous scale due to "richer contextual information." However, this analysis fails to de-confound this with sequence length. Later scales correspond to much longer sequences, which would naturally receive larger attention weights. This confounding factor is not excluded from the analysis.

6. The paper states that downsampling and upsampling are done via "interpolation" (Sec 3.1). What is the exact implementation of this? This detail is crucial for reproducibility.

7. Motif Scaffolding Centering: The paper mentions superimposing the ground-truth motif (Sec 4.2). This superimposition operation may change the center of the motif and the coordinate frame of the entire structure. How is this change in centering handled by the subsequent autoregressive steps and the flow decoder, which are sensitive to the absolute coordinate system?

8.  Since the flow-based decoder $v_{\theta}$ is shared across all scales, can a model trained on a 3-scale configuration be used for inference with a 5-scale configuration (or vice-versa)? How "agnostic" is the trained model to the number of scales used at inference time?

---

> ### Author Response · Authors · 2025-11-25
> **Official Comment by Authors: Quantitative Advantage of PAR**
>
> Thank you for your constructive feedback and thorough reviews. We truly appreciate your comment that "this is a **high-quality, well-written, and clear paper**" and your feedback that the paper demonstrates the new path for autoregressive modeling in protein structure generation. We have greatly improved our manuscript and addressed your concern as below. We sincerely thank you again and welcome any further feedback!
>
> > `W1:` The methodology leans heavily on a large, pre-existing flow-based model (Proteina) as its core decoder, which makes it difficult to assess the true contribution of the AR-only components. Given the marginal improvements over the decoder baseline, the added complexity of the AR framework is not fully justified. The paper feels more like an incremental application of an existing algorithm to a new domain rather than a fundamental breakthrough.
> >
>
> Thank you for raising this concern. We appreciate the opportunity to clarify the contributions of our AR component. In the following response, we demonstrate PAR's advantages in generation quality and sampling efficiency.
>
> **Generative performance:** In Table 1, PAR achieves *superior distribution-level FPSD performance* compared to all diffusion-based baselines except FrameDiff, which exhibits limited generation designability (65.4%). Beyond the initial results, we have explored two strategies to further unlock substantial improvements, leading to *state-of-the-art results* on both designability and FPSD metrics.
>
> **Unlock PAR's full potential on designability and FPSD.**
> We examined the experimental setup in our original submission, and have identified two reasons that caused the designability gap between PAR and Proteina. First, we didn't search for the best sampling hyperparameter $\gamma$ in our original experiments. By reducing the noise scaling parameter $\gamma$ from 0.45 to 0.3 in Equation 6 for the SDE sampling, we can reduce sampling stochasticity and improve sample quality, improving the designability from 88.0% to 96.00% without additional training.
>
> Second, we noticed that Proteina paper also reports the results from finetuned models on a PDB subset, which further improves FPSD while maintaining good designablity. Following their practice, we curate a PDB subset with 21K designable samples and finetune PAR on this subset for a brief 5k steps, after which PAR achieved:
>
> - Designability: increased from 88% to 96.60%
> - FPSD: increased from 231.50 to 160.99
> - **Outperforming all diffusion-based baselines on both metrics, even better than Proteina finetuned with the same PDB subset.**
>
> All new results are updated in Table 1.
>
> | Method | sc-rmsd < 2 (%) | sc-rmsd | Diversity | fid_pdb | fid_afdb | is_c | is_a | is_t | 2nd struct |
> | --- | --- | --- | --- | --- | --- | --- | --- | --- | --- |
> | PAR | 88.0 | 1.28 | **0.36** | 231.5 | **211.8** | 2.20 | 6.59 | 20.96 | 63.2 / 9.7 |
> | $\gamma=0.4$ | 91.00 | 1.18 | 0.37 | 256.23 | 237.69 | 2.21 | 6.63 | 19.62 | 65.9 / 8.7 |
> | $\gamma=0.35$ | 93.60 | 1.06 | 0.38 | 287.07 | 268.52 | 2.23 | 6.63 | 17.91 | 66.2 / 8.8 |
> | $\gamma=0.3$ | 96.00 | 1.01 | 0.39 | 313.86 | 296.40 | 2.24 | 6.60 | 16.71 | 66.3 / 8.9 |
> | Finetuning on PDB subset of 21k samples |  |  |  |  |  |  |  |  |  |  |
> | Proteina | 94.80 | 1.02 | 0.36 | 181.48 | 257.34 | **2.64** | 6.48 | **30.10** |  |
> | PAR | **96.60** | 1.04 | 0.43 | **160.99** | 228.44 | 2.57 | **7.42** | 23.61 | 50.2 / 16.7 |

---

> ### Author Response · Authors · 2025-11-25
> **Official Comment by Authors: Efficient Sampling**
>
> **Multi-scale orchestration of SDE/ODE for efficient sampling:** In our original submission, we did not take advantage of multi-scale model to improve the sampling efficiency. We use the same number of sampling steps (i.e., 1000 steps) at each scale for protein generation. With that said, we would like to highlight that there is actually an advantage for multi-scale model in terms of sampling efficiency. More specifically, (1) sampling at the coarser scale (e.g., first scale) is more efficient than sampling at finer scales (e.g., 2nd scale) due to shorter sequence length; (2) we can use less number of sampling steps at finer scales than coarser scales.
>
> Based on this rationale, we investigate how to leverage PAR's multi-scale design to improve sampling efficiency. By using SDE sampling only at the first scale, and switching to ODE sampling for the remaining scales, PAR could dramatically reduce the diffusion steps from 400 to 2 steps at the last two scales without harming designability. This is possible because a high-quality coarse topology places the model near high-density regions, enabling efficient refinement with ODE sampling alone. This is only achievable due to multi-scale design, and provides a clear efficiency advantage over single-scale models. We highlight several key findings:
>
> - **Recommended configuration: *SDE at the first scale + ODE at subsequent scales*.** Under this setup, the number of flow-matching steps at the last two scales can be reduced from 400 to 2 without decreasing designability (97%), yielding a 4.7x inference speedup. Crucially, SDE sampling at the first scale is necessary for establishing a reliable global topology.
> - **Why not reduce SDE steps?** Naively reducing the sampling steps significantly harms designability, dropping to 22% when reducing steps to 50, as shown in Figure 8. This is consistent with the observation of single-scale models like Proteina, where designability degrades to 89% when reducing SDE sampling steps to 200 (Table 2).
> - **Why not ODE everywhere?** ODE-only sampling exhibits poor designability (28%), confirming that SDE is essential at the coarse scale to explore global structure topology.
> - **Comparison to single-scale models:** Using its multi-scale formulation, PAR achieves **1.96x** and **2.5x** sampling speedup at length 150 and 200, respectively, compared to Proteina. This improvement is driven by speeding up the final scales, where the longer sequence lengths cause computational costs to grow quadratically in transformer architectures. Moreover, because the first scale size has a fixed size 64, the computational costs remain constant, even when generating longer sequences.
>
> We have included these new results in Fig. 8 and Table 2 in the revised PDF.
>
> **Inference speed analysis**
>
> |  |  | Length 150 |  | Length 200 |  |
> | --- | --- | --- | --- | --- | --- |
> | Sampling method | Sampling steps | Inference Time (s) | Designability (%) | Inference Time (s) | Designability (%) |
> | Proteina (SDE) | 0/0/400 | 131 | 97 | 170 | 92 |
> |  | 0/0/200 | 67 | 89 | 86 | 80 |
> | All SDE | 400/400/400 | 312 | 97 | 351 | 94 |
> |  | 400/400/2 | 184 | 0 |  |  |
> | All ODE | 400/400/400 | 312 | 28 |  |  |
> | S/S/O | 400/400/400 | 312 | 98 |  |  |
> |  | 400/400/2 | 184 | 99 | 186 | 91 |
> | S/O/O | 400/400/400 | 312 | 96 |  |  |
> |  | 400/2/2 | 67 | 97 | 68 | 94 |
>
> **Regarding incremental application:** We would like to emphasize that customizing an autoregressive formulation for protein structures entails unique challenges, making it more than a straightforward application of existing methods. Standard AR models typically rely on quantized latent space using tokenizers like RQ-VAE (e.g., VAR), which discards geometric fidelity due to discretization [1]. In contrast, our goal is to model structures directly in continuous data space, a setting where flow matching or diffusion naturally complements AR prediction. Removing the tokenizer, however, means we must redesign how protein structures are decomposed across scales, which motivates our multi-scale downsampling scheme in Sec. 3.1. Further, naively applying AR to 3D coordinates introduces severe exposure bias, which we found to dramatically degrade generation quality. To this end, we carefully mitigate this problem by introducing scheduled sampling and noisy context learning in Sec. 3.3. The use of a flow-based decoder is a practical way to leverage transformer architectures capable of modeling global dependencies, where its rationale is further clarified in our response to Q1. Importantly, the challenges above are intrinsic to AR models and would *not* be resolved by adopting a flow-based decoder alone.
>
> [1] Hsieh et al. Elucidating the design space of multimodel protein language models. ICML 2025.

---

> ### Author Response · Authors · 2025-11-25
>
> > `Q1:` **Reliance on Flow-Based Decoder**: The method is presented as an "autoregressive model," but it heavily relies on a large, **pre-trained** flow-based model (Proteina) as its decoder. This blurs the lines of novelty. It feels less like a pure AR model and more like a complex, multi-scale conditioning scheme for a flow model.
> >
>
> We would like to clarify that our motivation was specifically to explore autoregressive modeling in protein generation that could appropriately model the coarse-to-fine generation process. All the design choices are made to overcome limitations of standard AR models.
>
> **How to build AR models for protein structure?** Besides the limitations of unidirectional dependency and exposure bias, standard AR models may encounter fidelity loss that is introduced when we discrete 3D structures into tokens. We in turn build PAR directly in the data space, where the AR component followed by diffusion-based method becomes a natural choice.
>
> **Transformer-based decoder vs per-token decoder:** In selecting a decoder, our goals were not to rely on an existing structure decoder, but rather to use an architecture capable of *modeling global dependencies* needed for reliable coarse-to-fine predictions. In our preliminary study, we explored using a 3-layer MLP as the decoder head following MAR, as detailed in our response to Reviewer SoNE in Q1. However, MLP decoder failed to generate reasonable structures. This occurs because a per-token decoder is not expressive enough to capture the global correlations between atoms, which is required to produce a reliable coarse structure at the first scale for the subsequent coarse-to-fine refinement. Instead, utilizing transformer-based architectures offers the necessary capacity. However, the decoder in PAR is modular and can be replaced by alternative transformer-based models, e.g., DiT.
>
> **Adapting flow-based decoder:** On the architectural side, we used Proteina only for its transformer architecture and engineering support for flow-matching. We discarded many default setups used in protein structure models such as triangle layers, pair representations, and distogram-based auxiliary loss. We do this to ensure that the decoder is a minimal transformer implementation, which stands a strong distinction from the original Proteina.
>
> > **`Q2`: Questionable Benefit**: Given the relatively limited improvement over the Proteina baseline (Table 1), what is the practical benefit of introducing this complicated multi-scale AR scheme? The experiments do not convincingly demonstrate that this hybrid AR-flow approach is substantially better than the flow-based model it is built upon.
> >
>
> Thanks for pointing this out. We have identified the prior limitations of PAR and addressed them in our response to W1. In summary, PAR offers a clear advantage in terms of generation quality (designability), distribution-level modeling (FPSD), and sampling efficiency (speed).
>
> > `Q3:` **Weaker Designability (sc-RMSD)**: While the model excels at global fold distribution (FPSD), its fine-grained designability (sc-RMSD) is a weakness. In Table 1, the PAR (400M) model's sc-RMSD of 1.28 is comparable to Proteina (1.09).
> >
>
> Thank you for highlighting this. We have addressed this limitation in our response to W1. PAR now achieves substantial improvements in both FPSD and designability.
>
> > `Q4:` For evaluating diversity, the average pairwise TM-score (Table 1) is not very discriminative. A more robust metric, such as reporting the number of Foldseek clusters (as done in the Proteina paper), would be more convincing.
> >
>
> We appreciate this constructive suggestion. We have uploaded the results in the latest response.
>
> > `Q5:` The interpretation of the attention maps (Fig. 6) concludes that each scale attends to the previous scale due to "richer contextual information." However, this analysis fails to de-confound this with sequence length. Later scales correspond to much longer sequences, which would naturally receive larger attention weights. This confounding factor is not excluded from the analysis.
> >
>
> Thank you for pointing out this crucial detail. The attention score from each scale is first *averaged* over all residues within the scale. We then normalize them so that the attention scores across scales sum to 1. This eliminates the confounding factor that later scales might naturally receive larger attention weights. We have clarified this process in Sec. 4.3.

---

> > ### Author Response · Authors · 2025-11-25
> >
> > > `Q6:` The paper states that downsampling and upsampling are done via "interpolation" (Sec 3.1). What is the exact implementation of this? This detail is crucial for reproducibility.
> > >
> >
> > We use the nn.functional.interpolate function in Pytorch for both downsampling and upsampling:
> >
> > - For downsampling, we use "linear" mode:
> >     - `downsampled_structure = F.interpolate(raw_structure, size=size_at_curr_scale, mode='linear')`
> > - For upsampling from scale 1 to scale 2, we first upsample the prediction from scale 1 to full resolution, then apply downsampling:
> >     - `pred_raw_structure = F.interpolate(scale1_pred.unsqueeze(-1), size=(raw_length,1), mode='bicubic').squeeze(-1)`
> >     - `scale2_input = F.interpolate(pred_raw_structure, size=size_at_scale_2, mode='linear')`
> >     - This process iterates over scales.
> >
> > We are happy to further clarify any detail.
> >
> > >  `Q7:` Motif Scaffolding Centering: The paper mentions superimposing the ground-truth motif (Sec 4.2). This superimposition operation may change the center of the motif and the coordinate frame of the entire structure. How is this change in centering handled by the subsequent autoregressive steps and the flow decoder, which are sensitive to the absolute coordinate system?
> > >
> >
> > We superimpose the ground-truth motif and the corresponding motif segments on the model-predicted structure. The resulting rotation and translation from this alignment are applied to the ground-truth motif, and its transformed 3D coordinates are used to replace the motif segments on the predicted structures. Importantly, this process does not change the coordinate frame of the predicted structure and only alters the coordinates of the motif segments. After replacement, the integrated structure remains a coherent global topology, allowing the subsequent autoregressive steps to refine the details and produce the final structure.
> >
> > > `Q8:` Since the flow-based decoder is shared across all scales, can a model trained on a 3-scale configuration be used for inference with a 5-scale configuration (or vice-versa)? How "agnostic" is the trained model to the number of scales used at inference time?
> > >
> >
> > Thank you for bringing up this insightful point. In our original setup, we included a learnable scale embedding vector as part of the AR module's conditioning. This embedding allows the model to identify the current scale and adjust its behavior (e.g., generating coarse vs. fine structures). However, since the dimensionality of this learnable embedding is fixed to the number of scales, the model cannot be applied to a different scale configuration at inference.
> >
> > To explore flexible scale configurations, we finetune an alternative model that simply discards the learnable embedding on the PDB designable subset for 5k steps. This formulation cancels the embedding from a fixed number of scales and enables inference across arbitrary scale settings.
> >
> > As shown in the table below, when inferring with five scales using this 3-scale model, FPSD remains stable, suggesting that the model still captures the underlying data distribution under altered scale configurations. However, the designability substantially drops, indicating that sampling with an unseen scale configuration fails to preserve structural detail, ultimately leading to lower-quality results.
> >
> > |  | Designability |  | FPSD |  |  |
> > | --- | --- | --- | --- | --- | --- |
> > |  | (%) | sc-RMSD | vs. PDB | vs. AFDB | fs (C/A/T) |
> > | PAR (3 scale) | 96.6 | 1.04 | 160.99 | 228.44 | 2.57/7.42/23.61 |
> > | w/o scale emb | 92.8 | 1.16 | 175.09 | 246.34 | 2.54/7.66/26.68 |
> > | 5 scale inference | 72.6 | 1.74 | 177.01 | 246.76 | 2.56/7.53/26.78 |

---

> > > ### Comment · Reviewer_Zpd8 · 2025-11-26
> > >
> > > Thanks for the authors' detailed response and additional results. It is great to see the improved results of PAR, which clearly improve the quality of the paper. Most of my concerns are addressed in the authors' response.
> > >
> > > However, my concerns about the reliance on a heavy flow-based decoder remain. For this, I appreciate the authors share the thinking process behind the choice of decoder. However, as the authors said, a simple MLP decoder may not be expressive enough for generating good structures. That's why PAR still relies on a more expressive flow-based decoder. So instead of presenting PAR as "the first multi-scale autoregressive framework", I'd rather see it as a combination of autoregressive and flow-based methods for mutli-scale generation. Then, it is only interesting if we see significnat performance or speed improvement with this hybrid method.
> > >
> > > Overall, I still think this is a good paper worth acceptance. Though it is not as exciting as developing a new autoregressive framework for protein structure generation, the paper reveals some new insights and interesting results. I'll keep my score as 6.

---

> ### Author Response · Authors · 2025-11-27
>
> Thank you again for your thoughtful reviews and positive feedback of our work. We sincerely appreciate your detailed feedback, which has helped us greatly improved both the quality and clarity of the manuscript.
>
> > **AR with Flow Decoder**
>
> Regarding the concern about reliance on the flow-based decoder, we would like to explain why a high-capacity structure decoder is desirable even within an AR framework, especially in developing the first scale-wise protein autoregression in this work.
> The core generative process in PAR is driven by autoregressive approach: next-scale predictions are conditioned strictly on previously generated coarse-scale predictions. The flow-based decoder augments this framework to achieve the precision demand of protein modeling, but it does not replace the autoregressive process.
>
> The current adoption for flow-based decoder is largely due to the *precision* required in protein modeling in *atom-coordinate space* [1, 2].
> For instance, with an already strong representation encoder (Pairformer, ~150M parameters), AlphaFold3 still incorporates a ~300M-parameter diffusion-based structure module to achieve the necessary structural fidelity.
> Likewise, a sufficiently expressive decoder is beneficial for generation quality when paired with our autoregressive encoder, coherent to the best practices of atom-based protein generation. Meanwhile, we remove triangle layers and pair representations, achieving a lighter design that reduces memory consumption by 13× while still attaining strong generative performance.
>
>
> In addition, we discuss promising next steps towards building a pure AR structure model to reduce the need for heavy decoder.
>
>
> > **Towards Pure AR and Lightweight Decoder**
>
> In this work, we have effectively addressed many challenges of adapting AR models for protein structure, which stem from exposure bias and limited data (See Response to Q1 of Reviewer SoNE), paving the way to the development of autoregressive generation in protein design. We believe that the goal of building a pure AR model with lightweight diffusion head, even a minimal per-token MLP diffusion head, is achievable as the the next promising step.
>
> On one hand, training PAR within compact and expressive latent space (just like VAR and MAR) along with a highly-capable structure AE/VAEs might enable us to reduce the need for heavy decoder (required by data-space atomic modeling), following MAR. Many promising enheavors have been developed to create more powerful autoencoders trying to address the challenge such as information loss while preserving the generation quality [3]. We anticipate the advances of this promising direction, enpowering the paradigm of protein autoregressive modeling to achieve better generative capabilities.
>
> In addition, our analysis reveals that at all scales beyond the first, **only 2 ODE steps** are sufficient to achieve strong generative performance. We suggest that replacing the heavy flow decoder with a lightweight per-token MLP at finer scales is possible. Our rationale is that the global dependency has been well captured with the first-scale decoder and the AR transformer, refinement of structure details thus becomes less complex and could potentially be achieved with per-token decoder. This hybrid approach greatly reduces complexity of running heavy decoder across all scales.
>
> If this lightweight decoder strategy proves effective, the central modeling challenge reduces to **producing a strong first-scale structure**, which might be achievable by treating the AR transformer as a multi-task module that performs flow-matching to produce the first-scale structure and produces scale-wise condition for subsequent autoregressive steps [4].
>
> We warmly welcome any additional feedback you may have and would be excited to further refine the work based on your insights.
>
> [1] Geffner et al. Proteina. ICLR 2025. \
> [2] Abramson et al. AlphaFold3. Nature 2024. \
> [3] Xie et al. RAE. arXiv 2025. \
> [4] Zhou et al. Transfusion. arXiv 2024.

---

> > ### Author Response · Authors · 2025-12-03
> >
> > > `Q4:` For evaluating diversity, the average pairwise TM-score (Table 1) is not very discriminative. A more robust metric, such as reporting the number of Foldseek clusters (as done in the Proteina paper), would be more convincing.
> >
> > Thank you for your suggestions. We investigated the foldseek cluster diversity of PAR-generated samples. A larger $\gamma$ increases sampling stochasticity and leads to better diversity, reaching its peak value at $\gamma=0.6$. We generate 500 structures, with 100 samples for each length in $\{50, 100, 150, 200, 250\}$. We have updated the result in the Sec. C.7 of our paper.
> >
> > | $\gamma$ | Designable Clusters |
> > |-------| :-------------------: |
> > | 0.35  | 118  |
> > | 0.40   | 125     |
> > | 0.45  | 141   |
> > | 0.50   | 139   |
> > | 0.60   | **163**  |
> > | 0.70   | 159     |
> > | 0.80   | 145   |

---

### Official Review · Reviewer_SoNE · 2025-10-26

**Soundness:** 3
**Presentation:** 3
**Contribution:** 3
**Rating:** 4
**Confidence:** 3

**Summary:**

This paper introduces Protein Autoregressive Modeling (PAR), a novel multi-scale, coarse-to-fine framework for protein backbone generation.

**Key Contributions:**

*   **Hierarchical Generation:** PAR generates structures by first creating a coarse topology and then refining it over multiple scales, a process analogous to sculpting.
*   **Novel Architecture:** The model consists of three core components: (i) multi-scale downsampling operations, (ii) an autoregressive transformer for encoding information, and (iii) a flow-based backbone decoder.
*   **Addresses Exposure Bias:** The authors mitigate a common issue in autoregressive models by employing noisy context learning and scheduled sampling, leading to more robust generation.
*   **Strong Zero-Shot Generalization:** PAR can perform tasks like human-prompted conditional generation and motif scaffolding *without* needing to be fine-tuned, showcasing its flexibility and power.

**Strengths:**

### Strengths:

1.  The paper introduces a highly innovative approach by combining a **multi-scale autoregressive framework** with a **diffusion-based decoder** for protein backbone generation. This method allows for generating structures in a continuous coordinate space in a multi-scale manner.
2.  The practical advantages of the multi-scale autoregressive model are compellingly demonstrated through the *"Backbone generation with human prompt"* and *"Zero-shot motif scaffolding"* applications. These examples effectively showcase the model's flexibility and its capacity for precise, user-guided generation.

**Weaknesses:**

1.  **Lack of Analysis on Sampling Efficiency and Computational Cost:** A notable omission is the lack of analysis on sampling latency. While autoregressive (AR) models often present an advantage in generation speed over diffusion models in other domains, the PAR framework's architecture raises concerns. The model culminates in a diffusion decoder that appears computationally intensive (e.g., 1000 steps compared to Proteina's 400), and the preceding multi-scale AR stages introduce further computational overhead. This leads to a critical question: is PAR significantly slower than comparable end-to-end diffusion methods? Without this characterization, it is difficult to assess the practical trade-offs and viability of the proposed approach.

2.  **Unclear Justification for the Autoregressive Framework's Contribution to Performance:** While the multi-scale AR framework is conceptually elegant, its empirical necessity is not fully substantiated by the results. The model introduces considerable architectural complexity, yet this does not translate into superior performance. In fact, as shown in *Table 1*, the model underperforms a strong diffusion baseline like Proteina on the key metric of Designability, while showing no significant advantages in other reported metrics. Moreover, the framework pairs a powerful diffusion decoder with a relatively low-parameter AR encoder, and the scaling experiments primarily seem to reflect the scaling properties of the diffusion component. This makes it difficult to disentangle the contributions of the novel AR framework from the strong performance of the underlying decoder. The manuscript would be substantially strengthened by an ablation study or analysis that clearly demonstrates the value added by the AR mechanism.

3.  **Limited Demonstration on Longer Protein Chains:** The model's generalization capability is only demonstrated on proteins with lengths up to 256 residues, reflecting the scope of its training data. The generation of longer, multi-domain proteins is a crucial and challenging frontier for protein design. The performance of AR models, in particular, can degrade over long sequences due to error propagation. The paper would be more impactful if it included experiments on longer proteins or at least provided a thorough discussion of the potential challenges and limitations of scaling the PAR framework to these more complex and biologically relevant structures.

**Questions:**

1.  Regarding the model's architecture, the current design utilizes a low-parameter autoregressive (AR) encoder with a powerful diffusion decoder that generates all tokens for a given scale simultaneously. Have the authors considered or experimented with an alternative architecture, perhaps more aligned with models like **MAR**, which employs a large-parameter AR model paired with a lightweight decoder that generates tokens sequentially (i.e., per-token) at each scale? Such a design might better leverage the known strengths of AR models in sequence modeling.

2.  The proposed multi-scale framework relies on downsampling and upsampling operations that appear to be based on local windows in the 1D amino acid sequence. However, protein structures are defined by complex 3D relationships where residues that are distant in the 1D sequence can be close neighbors in 3D space. How does the current sequence-based sampling strategy ensure that these crucial non-local spatial relationships are effectively captured and preserved across different scales? Is there a risk of losing this vital information during the downsampling process?

3.  The generation of protein backbones occurs directly in the continuous and high-dimensional coordinate space. In other domains, such as image generation, autoregressive modeling directly on the raw data space (e.g., pixel space) is often less effective than modeling in a compressed latent space (e.g., from a VAE or VQ-VAE), as latent spaces can be more dense and numerically stable. Could the authors comment on this choice? What is their perspective on the challenges of applying autoregressive generation directly to the sparse and numerically sensitive coordinate space of proteins, and was a latent-space generation approach considered?

---

> ### Author Response · Authors · 2025-11-25
> **Official Comment by Authors: Inference Speed Analysis**
>
> Thank you so much for your constructive suggestions and for your positive feedback that "The paper introduces a highly innovative approach" and for recognizing the model's practical advantages demonstrated through zero-shot experiments. We sincerely appreciate your thoughtful reviews and address your concern below.
>
> > **`W1:` Lack of Analysis on Sampling Efficiency and Computational Cost:** A notable omission is the lack of analysis on sampling latency. While autoregressive (AR) models often present an advantage in generation speed over diffusion models in other domains, the PAR framework's architecture raises concerns. The model culminates in a diffusion decoder that appears computationally intensive (e.g., 1000 steps compared to Proteina's 400), and the preceding multi-scale AR stages introduce further computational overhead. This leads to a critical question: is PAR significantly slower than comparable end-to-end diffusion methods? Without this characterization, it is difficult to assess the practical trade-offs and viability of the proposed approach.
> >
>
> Thank you for raising the question about sampling efficiency. In our original submission, we did not take advantage of the multi-scale model and used the same number of sampling steps (i.e., 1000 steps) at each scale for protein generation. With that said, we would like to highlight that there is actually an advantage for multi-scale models in terms of sampling efficiency. More specifically, (1) sampling at the coarser scale (e.g., first scale) is more efficient than sampling at finer scales (e.g., 2nd scale) due to shorter sequence length; (2) we can use less number of sampling steps at finer scales than coarser scales.
>
> Based on this rationale, we have investigated how PAR's multi-scale design can be used to improve sampling efficiency, and we provide runtime analysis in Table 2 and Fig. 8. Our results show that PAR achieves a 2.5x sampling acceleration compared to the single-scale baseline, thereby properly leveraging the multi-scale design below.
>
> **Multi-scale orchestration of SDE/ODE for efficient sampling:** By using SDE sampling only at the first scale, and switching to ODE sampling for the remaining scales, PAR could dramatically reduce the diffusion steps from 400 to 2 steps at the last two scales without harming designability. This is possible because a high-quality coarse topology places the model near high-density regions, enabling efficient refinement with ODE sampling alone. This is only achievable due to multi-scale design, and provides a clear efficiency advantage over single-scale models. We highlight several key findings:
>
> - **Recommended configuration: *SDE at the first scale + ODE at subsequent scales*.** Under this setup, the number of flow-matching steps at the last two scales can be reduced from 400 to 2 without decreasing designability (97%), yielding a 4.7x inference speedup. Crucially, SDE sampling at the first scale is necessary for establishing a reliable global topology.
> - **Why not reduce SDE steps?** Naively reducing the sampling steps significantly harms designability, dropping to 22% when reducing steps to 50, as shown in Figure 8. This is consistent with the observation of single-scale models like Proteina, where designability degrades to 89% when reducing SDE sampling steps to 200 (Table 2).
> - **Why not ODE everywhere?** ODE-only sampling exhibits poor designability (28%), confirming that SDE is essential at the coarse scale to explore global structure topology.
> - **Comparison to single-scale models:** Using its multi-scale formulation, PAR achieves 1.96x and 2.5x sampling speedup at length 150 and 200, respectively, compared to Proteina. This improvement is driven by speeding up the final scales, where the longer sequence lengths cause computational costs to grow quadratically in transformer architectures. Moreover, because the first scale size has a fixed size 64, the computational costs remain constant, even when generating longer sequences.
>
> We have included these new results in Fig. 8 and Table 2 in the revised PDF.
>
> **Inference speed analysis**
>
> |  |  | Length 150 |  | Length 200 |  |
> | --- | --- | --- | --- | --- | --- |
> | Sampling method | Sampling steps | Inference Time (s) | Designability (%) | Inference Time (s) | Designability (%) |
> | Proteina (SDE) | 0/0/400 | 131 | 97 | 170 | 92 |
> |  | 0/0/200 | 67 | 89 | 86 | 80 |
> | All SDE | 400/400/400 | 312 | 97 | 351 | 94 |
> |  | 400/400/2 | 184 | 0 |  |  |
> | All ODE | 400/400/400 | 312 | 28 |  |  |
> | S/S/O | 400/400/400 | 312 | 98 |  |  |
> |  | 400/400/2 | 184 | 99 | 186 | 91 |
> | S/O/O | 400/400/400 | 312 | 96 |  |  |
> |  | 400/2/2 | 67 | 97 | 68 | 94 |

---

> ### Author Response · Authors · 2025-11-25
> **Official Comment by Authors: Quantitative Advantage of PAR**
>
> > **`W2:` Unclear Justification for the Autoregressive Framework's Contribution to Performance:** While the multi-scale AR framework is conceptually elegant, its empirical necessity is not fully substantiated by the results. The model introduces considerable architectural complexity, yet this does not translate into superior performance. In fact, as shown in *Table 1*, the model underperforms a strong diffusion baseline like Proteina on the key metric of Designability, while showing no significant advantages in other reported metrics. Moreover, the framework pairs a powerful diffusion decoder with a relatively low-parameter AR encoder, and the scaling experiments primarily seem to reflect the scaling properties of the diffusion component. This makes it difficult to disentangle the contributions of the novel AR framework from the strong performance of the underlying decoder. The manuscript would be substantially strengthened by an ablation study or analysis that clearly demonstrates the value added by the AR mechanism.
>
>
> Thanks for pointing out this important concern. PAR offers clear advantages in modeling the underlying data distribution. In Table 1, PAR achieves superior distribution-level FPSD performance compared to all diffusion-based baselines except FrameDiff, which exhibits limited generation designability (65.4%). Beyond the initial results, we have explored two strategies to further unlock substantial improvements, leading to **state-of-the-art results on both designability and FPSD metrics**.
>
> **Unlock PAR's full potential on designability and FPSD.** We examined the experimental setup in our original submission, and have identified two reasons that caused the designability gap. First, we didn't search for the best sampling hyperparameter $\gamma$ in our original experiments. By reducing the noise scaling parameter $\gamma$ from 0.45 to 0.3 in Equation 6 for the SDE sampling, we can reduce sampling stochasticity and improve sample quality, improving the designability from 88.0% to 96.00% without additional training.  Second, we noticed that Proteina paper also reports the results from finetuned models on a PDB subset, which further improves FPSD while maintaining good designablity. Following their practice, we curate a PDB subset with 21K designable samples and finetune PAR on this subset for a brief 5k steps, after which PAR achieved:
>
> - Designability: increased from 88.0% to 96.60%
> - FPSD: increased from 231.5 to **160.99**
> - **Outperforming all diffusion-based baselines on both metrics, even better than Proteina finetuned with the same PDB subset.**
>
> All new results are updated in Table 1.
> | Method | sc-rmsd < 2 (%) | sc-rmsd | Diversity | fid_pdb | fid_afdb | is_c | is_a | is_t | 2nd struct |
> | --- | --- | --- | --- | --- | --- | --- | --- | --- | --- |
> | PAR | 88.0 | 1.28 | **0.36** | 231.5 | **211.8** | 2.20 | 6.59 | 20.96 | 63.2 / 9.7 |
> | $\gamma=0.4$ | 91.00 | 1.18 | 0.37 | 256.23 | 237.69 | 2.21 | 6.63 | 19.62 | 65.9 / 8.7 |
> | $\gamma=0.35$ | 93.60 | 1.06 | 0.38 | 287.07 | 268.52 | 2.23 | 6.63 | 17.91 | 66.2 / 8.8 |
> | $\gamma=0.3$ | 96.00 | 1.01 | 0.39 | 313.86 | 296.40 | 2.24 | 6.60 | 16.71 | 66.3 / 8.9 |
> | Finetuning on PDB subset of 21k samples |  |  |  |  |  |  |  |  |  |  |
> | Proteina | 94.80 | 1.02 | 0.36 | 181.48 | 257.34 | **2.64** | 6.48 | **30.10** |  |
> | PAR | **96.60** | 1.04 | 0.43 | **160.99** | 228.44 | 2.57 | **7.42** | 23.61 | 50.2 / 16.7 |
>
> **Benefits of AR over single-scale diffusion decoder**: As shown above and in our response to W1, the multi-scale AR design improves both performance and efficiency. After finetuning, PAR surpasses the single-scale baseline (Proteina) in generative performance like designability and FPSD, and its multi-scale SDE/ODE orchestration yields a 2.5× sampling speedup.

---

> ### Author Response · Authors · 2025-11-25
> **Official Comment by Authors: Long Protein**
>
> > **`W3:` Limited Demonstration on Longer Protein Chains:** The model's generalization capability is only demonstrated on proteins with lengths up to 256 residues, reflecting the scope of its training data. The generation of longer, multi-domain proteins is a crucial and challenging frontier for protein design. The performance of AR models, in particular, can degrade over long sequences due to error propagation. The paper would be more impactful if it included experiments on longer proteins or at least provided a thorough discussion of the potential challenges and limitations of scaling the PAR framework to these more complex and biologically relevant structures.
> >
>
> Thank you for this insightful suggestion. Long-protein generation is indeed a challenging frontier for protein design, and we have included this experiment below to provide more insights.
>
> **Finetuning on longer protein chains.** We follow Proteina to finetune our models on datasets with longer proteins. Since Proteina has not released its long-protein dataset and its statistics, we cannot entirely reproduce their experiment setups. Instead, we follow the filtering procedure described in their appendix on PDB structures to curate a long-protein dataset. We filter PDB structures to lengths between 256 and 768 residues and keep only designable samples, resulting in ~26k high-quality proteins. The length-distribution of this dataset (Fig. 9) exhibits a long-tail shape with peaks around 300-400 residues. After ligand removal, 6.7% of the samples fall below the 256-residue threshold. We then finetune the 400M PAR and Proteina models in Table 1 on this dataset for 10k steps.
>
> **Long-protein generation and oversampling.** We generate 100 proteins for each length in {300, 400, 500, 600, 700}. PAR exhibits higher designability at lengths {300, 400}, consistent with the higher density of training samples in this range. At lengths between 500 to 700, both Proteina and PAR show degraded designability, while PAR demonstrating slightly better results. We attribute this to the long-tail nature of the training set, which includes far fewer samples in the length range between 500 and 700. The limited size of the training set (26K) also potentially hinders the model's from reaching its full potential. We leave scaling up long-protein data as a promising direction for future work.
>
> These results are included in Table 8 and Figure 9 in the revised PDF.
>
> |  | 300 |  | 400 |  | 500 |  | 600 |  | 700 |  |
> | --- | --- | --- | --- | --- | --- | --- | --- | --- | --- | --- |
> |  | sc-RMSD | Designability (%) | sc-RMSD | Designability (%) | sc-RMSD | Designability (%) | sc-RMSD | Designability (%) | sc-RMSD | Designability (%) |
> | Proteina | 1.91 | 85 | 2.70 | 61 | 4.09 | 49 | 7.90 | 21 | 13.32 | 4 |
> | PAR | **1.28** | **93** | **1.65** | **72** | **3.19** | **52** | **6.80** | **29** | **11.29** | **10** |

---

> > ### Author Response · Authors · 2025-11-25
> >
> > > **`Q1:`** Regarding the model's architecture, the current design utilizes a low-parameter autoregressive (AR) encoder with a powerful diffusion decoder that generates all tokens for a given scale simultaneously. Have the authors considered or experimented with alternative architecture, perhaps more aligned with models like MAR, which employs a large-parameter AR model paired with a lightweight decoder that generates tokens sequentially (i.e., per-token) at each scale? Such a design might better leverage the known strengths of AR models in sequence modeling.
> >
> > Thanks for bringing up this discussion. We introduced an ablation study examining the AR encoder size. We summarize key findings below.
> >
> > **Per-token vs per-scale decoder.** In our preliminary study, we implemented the model with a 200M-parameter AR module and, following MAR, used a 3-layer MLP (~20M) as the diffusion head. However, this setup failed to generate reasonable structures, yielding an average sc-RMSD of 16. This likely occurs because a per-token decoder is not expressive enough to capture the global correlations between atoms that is required to produce a reliable coarse structure at the first scale, which is crucial for the subsequent coarse-to-fine refinement. These observations motivated our shift to a per-scale transformer-based decoder.
> >
> > **Large vs. small decoder.** As shown in this table and our scaling experiments in Sec. 4.3, using a large decoder brings effective improvements to generation quality.
> >
> > **Large AR vs small AR.** With the decoder size fixed, increasing the AR transformer size from 60M to 400M does not offer improvements. We believe this is due to exposure bias: the AR module overfits to ground truth context to stabilize training, resulting in a mismatch with inference, where the model relies on its predictions as context.
> > This issue becomes more severe under several conditions:
> > 1. Larger AR models tend to overfit the context more strongly, making exposure bias more severe.
> > 2. Limited data increases overfitting risks: our ~588K training structures (32–256 residues each) provide far less coverage than datasets like ImageNet (1.28M 256x256 images)
> > 3. High precision tasks like protein modeling are sensitive to small errors, making exposure bias more serious than in image generation, where the compressed VAE latents lie in a smoother Gaussian space that is robust to small errors at the cost of some visual details [1,2].
> > Our noisy context learning and scheduled sampling mitigate this issue for the 60M PAR, but scaling the AR transformer appears to intensify this issue. Exploring more training data is a potential solution and we leave this for future work.
> >
> > We have included the results and the discussion above in **Sec. C.5** in the revised PDF.
> >
> > [1] Xie et al. RAE. arXiv 2025.
> > [2] He et al. JIT. arXiv 2025.
> >
> > |  | | Average |  |
> > | --- | --- | --- | --- |
> > | AR | Transformer Decoder | sc-RMSD | Designability (%) |
> > | 400M | 60M | 1.26 | 87.80 |
> > | 60M | 400M | 1.0101 | 96.00 |
> > | 60M | 60M | 1.19 | 92.60 |

---

> ### Author Response · Authors · 2025-11-25
> **Official Comment by Authors: Sequence-based Downsampling**
>
> > **`Q2:`** The proposed multi-scale framework relies on downsampling and upsampling operations that appear to be based on local windows in the 1D amino acid sequence. However, protein structures are defined by complex 3D relationships where residues that are distant in the 1D sequence can be close neighbors in 3D space. How does the current sequence-based sampling strategy ensure that these crucial non-local spatial relationships are effectively captured and preserved across different scales? Is there a risk of losing this vital information during the downsampling process?
>
> Thank you for raising this interesting point. At coarser scales, downsampling could discard some information of geometric details, which exactly aligns with our motivation that model should learn a coarse topology first before sculpting the local details.
>
> Meanwhile, an interesting question is whether 1D sequence downsampling properly preserves the 2D pairwise spatial relationships in protein structures. To study this, we attempt to investigate the difference between pairwise distances computed after downsampling the 1D coordinate sequence and those obtained by downsampling the full-resolution 2D distance map. We discuss details below.
>
>
> **Spatial relationships in downsampled 1D sequence.** We follow the process below to quantify the spatial relationships:
> 1. Downsample the coordinate sequence from $\mathbb{R}^{L\times3}$ to $\mathbb{R}^{size(i)\times3}$ for each scale i.
> 2. We compute pairwise distance maps using the downsampled sequence, leading to a $\texttt{size}(i) \times \texttt{size}(i)$ map.
>
> **Spatial relationships in 3D space after downsampling.** We quantify this using the pairwise distance map calculated from the full-resolution structure:
> 1. Calculate the pairwise distance map of the structure, producing a $L \times L$ map.
> 2. We downsample pairwise map this using the `F.interpolate(mode='bicubic')` operation, resulting in a $\texttt{size}(i) \times \texttt{size}(i)$ map.
>
> **Does sequence-based downsampling preserve spatial relationships?** We select all samples from the testing set, and calculate the RMSE and lddt between the aforementioned two $\texttt{size}(i) \times \texttt{size}(i)$ pairwise maps for each sample. As expected, rmse slightly increases as $\texttt{size}(i)$ decreases, reflecting the loss of fine-grained details at coarser scales. However, lddt remains consistently at 1 and the rmse values remain low across all scales. Together, these results indicate that, despiste small information loss at the coarse scales, 1D sequence downsampling preserves the essential pairwise spatial correlations captured by the downsampled 2D distance map.
>
> | Size(i) | 16 | 32 | 64 | 128 |
> | --- | --- | --- | --- | --- |
> | rmse | 0.362 | 0.275 | 0.2172 | 0.170 |
> | lddt | 1 | 1 | 1 | 1 |
>
>
>
> We have included this discussion in Sec. C.6.

---

> > ### Author Response · Authors · 2025-11-25
> >
> > > **`Q3:`** The generation of protein backbones occurs directly in the continuous and high-dimensional coordinate space. In other domains, such as image generation, autoregressive modeling directly on the raw data space (e.g., pixel space) is often less effective than modeling in a compressed latent space (e.g., from a VAE or VQ-VAE), as latent spaces can be more dense and numerically stable. Could the authors comment on this choice? What is their perspective on the challenges of applying autoregressive generation directly to the sparse and numerically sensitive coordinate space of proteins, and was a latent-space generation approach considered?
> >
> > We discuss our rationales below.
> >
> > **Why not VQ-VAE?** Discretization with VQ-VAE generally harms generative performance. In image generation, autoregressive models commonly rely on tokenizers to discretize images into tokens [1]. For proteins, however, discretizing 3D structures into tokens leads to substantial fidelity loss and severely degrades generative performance. This was demonstrated by Hsieh et al. 2025 [2]. To avoid this issue, we instead leverage flow-based decoder to bypass tokenization.
> >
> > **Why not VAE?**  VAEs are popular in image generation because images contain high redundancy, allowing compression into low-dimensional latents to enable efficient modeling [3]. Protein structures, however, demand high accuracy, and compressing these structures risks losing geometric details or fidelity. To our knowledge, latent-based approaches often fail to deliver competitive performance [4]. A promising direction is hybrid modeling in both latent and data spaces. For instance, La-Proteina [5] models Ca coordinates in the data space and uses a VAE decoder to further achieve joint modeling of sequence and all-atom coordinates. In our case, we focus on modeling Ca atoms. Direct modeling in the data space hence remains the simplest and most effective option.
> >
> > **Challenges of applying autoregressive generation to data space.** Aside from limitations of standard AR models that have been addressed in our paper, including exposure bias, unidirectional dependency and reliance on discretization, we note that learning rotation invariance embeddings is a broad challenge for data space modeling. Fortunately, batch multiplicity with rotation augmentation has been shown effective in AlphaFold3 [5] to tackle this challenge.
> >
> > [1] Tian et al. VAR. NeurIPS 2024. \
> > [2] Hsieh et al. Elucidating the Design Space of Multimodal Protein Language Models. ICML 2025. \
> > [3] Rombach et al. Latent Diffusion. CVPR 2022. \
> > [4] Fu et al. A latent diffusion model for protein structure generation. Learning on graphs conference 2024. \
> > [5] Geffner et al. La-Proteina. ArXiv 2025. \
> > [6] Abransom et al. AlphaFold3. Nature 2024.

---

### Official Review · Reviewer_8etu · 2025-10-30

**Soundness:** 3
**Presentation:** 3
**Contribution:** 3
**Rating:** 4
**Confidence:** 4

**Summary:**

This paper introduces Protein Autoregressive Modeling (PAR), the multi-scale autoregressive framework for protein backbone generation. PAR generates protein structures in a coarse-to-fine manner, progressively refining structural details across scales.
To address exposure bias and train–generation mismatches typical of autoregressive models, the authors introduce noisy context learning and scheduled sampling, enhancing robustness during backbone generation.
Empirically, PAR demonstrates strong zero-shot generalization, enabling conditional protein design and motif scaffolding without additional fine-tuning.

**Strengths:**

- The paper proposes the *first* multi-scale autoregressive model for protein backbone generation, integrating coarse-to-fine prediction within a single generative process.
- The introduction of *noisy context learning* and *scheduled sampling* provides a principled way to mitigate exposure bias and mismatch between training and inference, a common challenge in autoregressive models.
- PAR demonstrates strong *zero-shot generalization* and supports flexible conditional tasks such as motif scaffolding and human-guided protein design, showing versatility beyond unconditional generation.

**Weaknesses:**

-  Although the proposed multi-scale autoregressive formulation is conceptually novel, the paper does not clearly demonstrate *quantitative or qualitative advantages* over existing diffusion-based protein generative models in terms of either *generation quality* or *generation efficiency*.
- In the *zero-shot generalization* experiments, the paper mainly focuses on demonstrating conditional controllability (e.g., motif scaffolding) but does not evaluate the *designability* or *physical plausibility* of the generated structures. For motif scaffolding, comparisons with other *training-based scaffolding approaches* would be important to contextualize the results.
- The paper reports ablations only on length versus ratio, while Figure 2 uses five scales and Table 1 uses three. There is no clear analysis or justification for how the number of scales influences performance, which is central to the proposed multi-scale formulation.
- Since autoregressive and diffusion-based models have distinct sampling paradigms, *sampling time* is an important metric for practical evaluation. The paper does not report runtime comparisons or efficiency analyses, leaving uncertainty about the computational trade-offs of the proposed approach.

**Questions:**

Q1   Intuitively, providing a hierarchical coarse-to-fine prior should make the final generation easier. However, the reported results suggest that PAR underperforms compared to diffusion models that directly generate the final-scale backbone. Could the authors clarify why the multi-scale autoregressive design does not yield stronger advantages in practice?

Q2   The paper shows a case of zero-shot motif scaffolding involving a *continuous* motif segment. Can PAR also handle *discontinuous* or multi-segment motifs in a zero-shot setting? If not, what are the current limitations that prevent it?

Q3   How is scheduled sampling concretely implemented during training? Does it require running two forward passes per iteration? If self-conditioning is also used, does that effectively multiply the computational cost (e.g., four forward passes per iteration)?

---

> ### Author Response · Authors · 2025-11-25
> **Official Comments by Authors: Quantitative Advantage of PAR**
>
> Thank you so much for your constructive suggestions and we truely appreciate your comment that "PAR demonstrates versatility beyond unconditional generation". We have greatly improved our manuscript and addressed your concern as below. We sincerely thank you again and welcome any further feedback!
>
> > **`W1:`** Although the proposed multi-scale autoregressive formulation is conceptually novel, the paper does not clearly demonstrate *quantitative or qualitative advantages* over existing diffusion-based protein generative models in terms of either *generation quality* or *generation efficiency*.
>
> Thanks for pointing out this crucial discussion. PAR offers clear advantages in modeling the underlying data distribution. In Table 1, PAR achieves **superior distribution-level FPSD performance** compared to all diffusion-based baselines except FrameDiff, which exhibits limited generation designability (65.4%). Beyond the initial results, we have explored two strategies to further unlock substantial improvements, leading to **state-of-the-art results on both designability and FPSD metrics**. In addition, we extend our discussion on efficient sampling by dramatically reducing diffusion steps below.
>
> **Unlock PAR's full potential on designability and FPSD.**
> We examined the experimental setup in our original submission, and have identified two reasons that caused the designability gap. First, we didn't search for the best sampling hyperparameter $\gamma$ in our original experiments. By reducing the noise scaling parameter $\gamma$ from 0.45 to 0.3 in Equation 6 for the SDE sampling, we can reduce sampling stochasticity and improve sample quality, improving the designability from 88.0% to 96.00% without additional training.
>
> Second, we noticed that Proteina paper also reports the results from finetuned models on a PDB subset, which further improves FPSD while maintaining good designablity. Following their practice, we curate a PDB subset with 21K designable samples and finetune PAR on this subset for a brief 5k steps, after which PAR achieved:
>
> - Designability: increased from 88% to 96.60%
> - FPSD: increased from 231.50 to 160.99
> - **Outperforming all diffusion-based baselines on both metrics, even better than Proteina finetuned with the same PDB subset.**
>
> All new results are updated in Table 1.
>
> | Method | sc-rmsd < 2 (%) | sc-rmsd | Diversity | fid_pdb | fid_afdb | is_c | is_a | is_t | 2nd struct |
> | --- | --- | --- | --- | --- | --- | --- | --- | --- | --- |
> | PAR | 88.0 | 1.28 | **0.36** | 231.5 | **211.8** | 2.20 | 6.59 | 20.96 | 63.2 / 9.7 |
> | $\gamma=0.4$ | 91.00 | 1.18 | 0.37 | 256.23 | 237.69 | 2.21 | 6.63 | 19.62 | 65.9 / 8.7 |
> | $\gamma=0.35$ | 93.60 | 1.06 | 0.38 | 287.07 | 268.52 | 2.23 | 6.63 | 17.91 | 66.2 / 8.8 |
> | $\gamma=0.3$ | 96.00 | 1.01 | 0.39 | 313.86 | 296.40 | 2.24 | 6.60 | 16.71 | 66.3 / 8.9 |
> | Finetuning on PDB subset of 21k samples |  |  |  |  |  |  |  |  |  |  |
> | Proteina | 94.80 | 1.02 | 0.36 | 181.48 | 257.34 | **2.64** | 6.48 | **30.10** |  |
> | PAR | **96.60** | 1.04 | 0.43 | **160.99** | 228.44 | 2.57 | **7.42** | 23.61 | 50.2 / 16.7 |

---

> ### Author Response · Authors · 2025-11-25
> **Official Comments by Authors: Efficient Sampling with PAR**
>
> **Multi-scale orchestration of SDE/ODE for efficient sampling:** Thank you for raising the question about sampling efficiency. In our original submission, we did not take advantage of multi-scale model to improve the sampling efficiency. We use the same number of sampling steps (i.e., 1000 steps) at each scale for protein generation. With that said, we would like to highlight that there is actually an advantage for multi-scale model in terms of sampling efficiency. More specifically, (1) sampling at the coarser scale (e.g., first scale) is more efficient than sampling at finer scales (e.g., 2nd scale) due to shorter sequence length; (2) we can use less number of sampling steps at finer scales than coarser scales.
>
> Based on this rationale, we investigate how to leverage PAR's multi-scale design to improve sampling efficiency. By using SDE sampling only at the first scale, and switching to ODE sampling for the remaining scales, PAR could dramatically reduce the diffusion steps from 400 to 2 steps at the last two scales without harming designability. This is possible because a high-quality coarse topology places the model near high-density regions, enabling efficient refinement with ODE sampling alone. This is only achievable due to multi-scale design, and provides a clear efficiency advantage over single-scale models. We highlight several key findings:
>
> - **Recommended configuration: *SDE at the first scale + ODE at subsequent scales*.** Under this setup, the number of flow-matching steps at the last two scales can be reduced from 400 to 2 without decreasing designability (97%), yielding a 4.7x inference speedup. Crucially, SDE sampling at the first scale is necessary for establishing a reliable global topology.
> - **Why not reduce SDE steps?** Naively reducing the sampling steps significantly harms designability, dropping to 22% when reducing steps to 50, as shown in Figure 8. This is consistent with the observation of single-scale models like Proteina, where designability degrades to 89% when reducing SDE sampling steps to 200 (Table 2).
> - **Why not ODE everywhere?** ODE-only sampling exhibits poor designability (28%), confirming that SDE is essential at the coarse scale to explore global structure topology.
> - **Comparison to single-scale models:** Using its multi-scale formulation, PAR achieves 1.96x and 2.5x sampling speedup at length 150 and 200, respectively, compared to Proteina. This improvement is driven by speeding up the final scales, where the longer sequence lengths cause computational costs to grow quadratically in transformer architectures. Moreover, because the first scale size has a fixed size 64, the computational costs remain constant, even when generating longer sequences.
>
> We have included these new results in Fig. 8 and Table 1&2 in the revised PDF.
>
> **Inference speed analysis**
>
> |  |  | Length 150 |  | Length 200 |  |
> | --- | --- | --- | --- | --- | --- |
> | Sampling method | Sampling steps | Inference Time (s) | Designability (%) | Inference Time (s) | Designability (%) |
> | Proteina (SDE) | 0/0/400 | 131 | 97 | 170 | 92 |
> |  | 0/0/200 | 67 | 89 | 86 | 80 |
> | All SDE | 400/400/400 | 312 | 97 | 351 | 94 |
> |  | 400/400/2 | 184 | 0 |  |  |
> | All ODE | 400/400/400 | 312 | 28 |  |  |
> | S/S/O | 400/400/400 | 312 | 98 |  |  |
> |  | 400/400/2 | 184 | 99 | 186 | 91 |
> | S/O/O | 400/400/400 | 312 | 96 |  |  |
> |  | 400/2/2 | 67 | 97 | 68 | 94 |
>
> > **`W2:`** In the *zero-shot generalization* experiments, the paper mainly focuses on demonstrating conditional controllability (e.g., motif scaffolding) but does not evaluate the *designability* or *physical plausibility* of the generated structures. For motif scaffolding, comparisons with other *training-based scaffolding approaches* would be important to contextualize the results.
> >
> Thanks for your suggestions. We have uploaded the results in the latest response.

---

> ### Author Response · Authors · 2025-11-25
> **Official Comment by Authors: Scale Ablation Study**
>
> > **`W3:`** The paper reports ablations only on length versus ratio, while Figure 2 uses five scales and Table 1 uses three. There is no clear analysis or justification for how the number of scales influences performance, which is central to the proposed multi-scale formulation.
>
> Thanks for the suggestions. We include new results (bolded) below, including PAR with two, four, and five scales. We have updated these new results in Table 4. We used PAR with 60M parameters for this ablation study. PAR obtains better designability and FPSD when increasing from two scales to three scales. Beyond this point, increasing the scale configurations to four and five scales results in degraded designability, potentially due to error accumulation and exposure bias. These results support our choice of adopting the 3-scale PAR as the default.
>
> |  | Designability | sc-RMSD | FPSD vs. |  | fS (C/A/T) |
> | --- | --- | --- | --- | --- | --- |
> |  | (%) | (Å) | PDB | AFDB |  |
> | **{64, 256}** | 83.0 | 1.38 | 282.85 | 274.32 | 2.14 / 6.58 / 20.66 |
> | {64, 128, 256} | 85.0 | 1.39 | 279.63 | 267.35 | 2.15 / 6.52 / 20.35 |
> | **{64, 128, 192, 256}** | 77.8 | 1.55 | 296.70 | 282.69 | 2.05 / 6.04 / 18.69 |
> | **{64, 96, 128, 192, 256}** | 81.0 | 1.51 | 276.00 | 263.58 | 2.17 / 6.31 / 20.65 |
> | {L // 4, L // 2, L} | 86.4 | 1.49 | 310.64 | 298.30 | 2.00 / 5.87 / 18.91 |
>
> > **`W4:`** Since autoregressive and diffusion-based models have distinct sampling paradigms, sampling time is an important metric for practical evaluation. The paper does not report runtime comparisons or efficiency analyses, leaving uncertainty about the computational trade-offs of the proposed approach.
>
> Thanks for highlighting this practical point. We have added a sampling runtime analysis in Table 2. Please see our response to W1 for detailed discussion.
>
> > **`Q1:`** Intuitively, providing a hierarchical coarse-to-fine prior should make the final generation easier. However, the reported results suggest that PAR underperforms compared to diffusion models that directly generate the final-scale backbone. Could the authors clarify why the multi-scale autoregressive design does not yield stronger advantages in practice?
>
> Thanks for pointing this out. Initial results showed that PAR achieved stronger distribution-level FPSD than all single-scale diffusion baselines except FrameDiff, whose designability is substantially lower (65.4%), indicating that PAR models the underlying data distribution more effectively.
>
> **Unlock PAR's full potential on designability and FPSD.** In our response to W1, we have addressed this limitation via better noise scaling parameters and a simple data filtering mechanism. With these approaches, PAR now outperforms all diffusion baselines in terms of both FPSD (160.99) and designability (96.60%). We include new results in Table 1.
>
> **Hierarchical coarse-to-fine prior for efficient sampling.** We agree with the reviewer's intuition that "providing a hierarchical coarse-to-fine prior should make the final generation easier". PAR's performance and sampling behavior directly support this: the high-quality topology produced on the first scale provides a strong prior that enables efficient sampling. Using ODE sampling on later scales, we can reduce the flow-matching steps from 400 to just 2 without sacrificing designability. Typically, single-scale models like Proteina suffer substantial designability drops under ODE sampling and thus cannot achieve such efficiency gains without additional training. This new analysis is updated in Figure 8, Table 2, and detailed in our response to W1.
>
>
> > **`Q2:`** The paper shows a case of zero-shot motif scaffolding involving a continuous motif segment. Can PAR also handle discontinuous or multi-segment motifs in a zero-shot setting? If not, what are the current limitations that prevent it?
>
> Thanks for introducing this discussion. We have uploaded the results in the latest response. PAR is able to produce solutions on tasks (e.g., 1QJG, 2KL8) with discontinuous motif segments in a zero-shot manner.
>
> [1] RFDiffusion. Watson et al. Nature, 2023.

---

> > ### Author Response · Authors · 2025-11-25
> >
> > > **`Q3:`** How is scheduled sampling concretely implemented during training? Does it require running two forward passes per iteration? If self-conditioning is also used, does that effectively multiply the computational cost (e.g., four forward passes per iteration)?
> > >
> >
> > Thanks for bringing up this discussion. We clarify the details below:
> >
> > **Forward passes:**
> >
> > - AR transformer: always one forward pass per scale
> > - Flow-based decoder: one forward pass when self-conditioning is not used (50%), otherwise two (50%).
> >
> > **Scheduled sampling:**
> >
> > The scheduled sampling process highly resembles the inference process:
> >
> > - Scale 1: AR transformer and flow-based decoder predicts the structure $x^1_\text{pred}$, where the flow-based decoder might run two forward passes depending on self-conditioning.
> > - Subsequent scales: With a probability of 0.50, we replace the ground truth context $x^1$ with $x^1_\text{pred}$. We detach $x^1_\text{pred}$ before the replacement to avoid heavy gradient computation. If the ground truth context is replaced, the context becomes {\<bos\>, $x^1_\text{pred}$}. We then use AR transformer and flow-based decoder to obtain $x^2_\text{pred}$. Similarly, flow-based decoder runs one or two forward passes depending on self-conditioning.
> > - We sequentially sum the loss obtained from each scale.
> > - This process continues until all scales are processed.
> > - Training speed: Empirically, schedule sampling slightly increases the required training time by 20%, when tested with our 60M models trained on 8 gpus with a batch size of 15 for 100k steps.

---

> > > ### Author Response · Authors · 2025-12-03
> > >
> > > > `W2:` In the zero-shot generalization experiments, the paper mainly focuses on demonstrating conditional controllability (e.g., motif scaffolding) but does not evaluate the designability or physical plausibility of the generated structures. For motif scaffolding, comparisons with other training-based scaffolding approaches would be important to contextualize the results.
> > >
> > > We quantify the **zero-shot** motif scaffolding performance of PAR along with other training-based approaches. The table below reports the ratio of unique solutions divided by total generated samples.
> > >
> > > We use PAR to generate 100 backbone structures for each benchmark problem in [1]. Following Proteina's evaluation protocol, we produce 8 ProteinMPNN sequences for each generation with the motif residues fixed, and feed each sequence to ESMFold. Using the predicted structure, we calculate ca-RMSD and MotifRMSD. A design is considered a success if any sequence achieves scRMSD ≤ 2Å, a motifRMSD ≤ 1Å, pLDDT ≥ 70, and pAE ≤ 5. All successful samples are clustered via hierarchical clustering and a TM-score threshold of 0.6 to obtain the number of unique solutions. For other training-based methods, we directly quote the results reported in Proteina, which evaluate results using 1000 samples.
> > >
> > > We have added the results in the Sec. C.8 of our paper.
> > >
> > > [1] RFDiffusion. Watson et al. Nature, 2023.
> > >
> > > |                                 | Ours (100 samples) | Proteina | Genie2 | RFDiffusion | FrameFlow |
> > > |---------------------------------|:------------------:|:--------:|:------:|:-----------:|:---------:|
> > > | # tasks with >= 1% success rate |         11         |    11    |   11   |      9      |     11    |
> > > | 1PRW |          0         |     0.3    |    0.2   | 0.1 | 0.3 |
> > > | 1BCF |          0         |     0.1    |    0.1   | 0.1 | 0.1 |
> > > | 5TPN |          0         |     0.4    |    0.8   | 0.5      |  0.6     |
> > > | 5IUS |          0         |     0.1    |    0.1   | 0.1      |     0     |
> > > | 3IXT |          9.0         |     0.8    |   1.4   |      0.3      |     0.8     |
> > > | 5YUI |          0         |     0.5    |    0.3   | 0.1  |     0.1     |
> > > | 1QJG |          3.0         |     0.3    |    0.5   |      0.1      |     1.8    |
> > > | 1YCR |          4.0         |    24.9   |   134  |      0.7      |    14.9    |
> > > | 2KL8 |          4.0         |     0.1    |    0.1   |      0.1      |     0.1     |
> > > | 7MRX.60 |       0      |     0.2    |    0.5   |      0.1      |     0.1     |
> > > | 7MRX.85 |       1.0         |    3.1    |   2.3   |      1.3     |     2.2    |
> > > | 7MRX.128 |     1.0         |    5.1    |   2.7   |      6.6     |     3.5    |
> > > | 4JHW     |          0    |     0    |    0   |      0      |     0     |
> > > | 4ZYP     |          0         |    1.1    |    0.3   |      0.6      |     0.4     |
> > > | 5WN9     |          0         |     0.2    |    0.1   |      0      |     0.3     |
> > > | 5TRV_short |      0        |     0.1    |    0.3   |      0.1      |     0.1     |
> > > | 5TRV_med   |          0         |    2.2    |   2.3   |      1.0     |     2.1    |
> > > | 5TRV_long  |          0         |    17.9   |   9.7   |      2.3     |     7.7    |
> > > | 6E6R_short |          8.0         |    5.6    |   2.6   |      2.3     |     2.5    |
> > > | 6E6R_med  |          2.0         |    41.7   |   27.2  |     15.1     |     9.9    |
> > > | 6E6R_long |          3.0         |    71.3   |   41.5  |     38.1     |    11.0    |
> > > | 6EXZ_short |          2.0         |     0.3    |    0.2   |      0.1      |     0.3 |
> > > | 6EXZ_med  |          9.0         |    4.3    |   5.4   |      2.5     |    11.0    |
> > > | 6EXZ_long   |         12.0         |    29.0   |   32.6  |     16.7     |    40.3    |

---

### Official Review · Reviewer_Knjj · 2025-10-31

**Soundness:** 3
**Presentation:** 3
**Contribution:** 3
**Rating:** 6
**Confidence:** 5

**Summary:**

Protein Autoregressive Modeling (PAR) is a multi-scale framework for protein backbone generation. PAR generates the coarse structure first then refines the structural details. PAR obtains competitive results to prior single scale methods, while also being able to structurally scaffold proteins without any finetuning.

**Strengths:**

- High technical novelty. The ability to define different granularities of the coarse structure to then refine is applicable to design tasks
- Competitive performance. Demonstrates that the AR factorization of the problem works as desired without much loss of performance across any benchmarks.
-Clear concise well written.
- The fact that the AR process is done without explicit tokenization  driven through the CA flow loss is elegant.

**Weaknesses:**

- No formal motif benchmark. Even if not SOTA it would be interesting to see.
- No comparison of the inference speed

**Questions:**

- Do you use TriMul or just the attention pair bias Proteina architecture for the AR and flow components?
-If one were to run PAR with just scale {L} what happens? Table 3 seems like the results are slightly harmed by more downsampling but curious if you basically get Proteina in the limit of doing no downsampling.
- Does PAR hold to long length designability? Given its Proteina based the long length study could be extended here. it would be interesting to see if there is any value to being able to prompt a long length model.

---

> ### Author Response · Authors · 2025-11-25
> **Inference Speedup of PAR**
>
> Thank you for your insightful feedback. We appreciate you for recognizing the high technical novelty of our paper. We address your concerns below. We sincerely thank you once again and welcome any further feedback.
>
> > **`W1:` No formal motif benchmark. Even if not SOTA it would be interesting to see.**
>
> Thanks for your suggestions. We have uploaded the results in the latest response.
>
> [1] RFDiffusion. Watson et al. Nature, 2023.
>
> >**`W2:` Comparison of the inference speed**
>
> Thank you for raising the question about sampling efficiency. In our original submission, we did not take advantage of the multi-scale model and used the same number of sampling steps (i.e., 1000 steps) at each scale for protein generation. With that said, we would like to highlight that there is actually an advantage for multi-scale models in terms of sampling efficiency. More specifically, (1) sampling at the coarser scale (e.g., first scale) is more efficient than sampling at finer scales (e.g., 2nd scale) due to shorter sequence length; (2) we can use less number of sampling steps at finer scales than coarser scales.
> Based on this rationale, we have investigated how PAR's multi-scale design can be used to improve sampling efficiency, and we provide runtime analysis in Fig. 8 and Table. 2 (long protein). Our results show that PAR achieves a 2.5x sampling acceleration compared to the single-scale baseline, thereby properly leveraging the multi-scale design below.
>
> **Multi-scale orchestration of SDE/ODE for efficient sampling:** By using SDE sampling only at the first scale, and switching to ODE sampling for the remaining scales, PAR could dramatically reduce the diffusion steps from 400 to 2 steps at the last two scales without harming designability. This is possible because a high-quality coarse topology places the model near high-density regions, enabling efficient refinement with ODE sampling alone. This is only achievable due to multi-scale design, and provides a clear efficiency advantage over single-scale models. We highlight several key findings:
>
> - **Recommended configuration: *SDE at the first scale + ODE at subsequent scales*.** Under this setup, the number of flow-matching steps at the last two scales can be reduced from 400 to 2 without decreasing designability (97%), yielding a 4.7x inference speedup. Crucially, SDE sampling at the first scale is necessary for establishing a reliable global topology.
> - **Why not reduce SDE steps?** Naively reducing the sampling steps significantly harms designability, dropping to 22% when reducing steps to 50, as shown in Figure 8. This is consistent with the observation of single-scale models like Proteina, where designability degrades to 89% when reducing SDE sampling steps to 200 (Table 2).
> - **Why not ODE everywhere?** ODE-only sampling exhibits poor designability (28%), confirming that SDE is essential at the coarse scale to explore global structure topology.
> - **Comparison to single-scale models:** Using its multi-scale formulation, PAR achieves **1.96x** and **2.5x** sampling speedup at length 150 and 200, respectively, compared to Proteina. This improvement is driven by speeding up the final scales, where the longer sequence lengths cause computational costs to grow quadratically in transformer architectures. Moreover, because the first scale size has a fixed size 64, the computational costs remain constant, even when generating longer sequences.
>
> We have included these new results in Fig. 8 and Table 1&2 in the revised PDF.
>
> **Inference speed analysis**
>
> |  |  | Length 150 |  | Length 200 |  |
> | --- | --- | --- | --- | --- | --- |
> | Sampling method | Sampling steps | Inference Time (s) | Designability (%) | Inference Time (s) | Designability (%) |
> | Proteina (SDE) | 0/0/400 | 131 | 97 | 170 | 92 |
> |  | 0/0/200 | 67 | 89 | 86 | 80 |
> | All SDE | 400/400/400 | 312 | 97 | 351 | 94 |
> |  | 400/400/2 | 184 | 0 |  |  |
> | All ODE | 400/400/400 | 312 | 28 |  |  |
> | S/S/O | 400/400/400 | 312 | 98 |  |  |
> |  | 400/400/2 | 184 | 99 | 186 | 91 |
> | S/O/O | 400/400/400 | 312 | 96 |  |  |
> |  | 400/2/2 | 67 | 97 | 68 | 94 |

---

> ### Author Response · Authors · 2025-11-25
> **Official Comment by Authors: Long Protein and Architecture Clarifications**
>
> >**`Q1:`** Do you use TriMul or just the attention pair bias Proteina architecture for the AR and flow components? -If one were to run PAR with just scale {L} what happens? Table 3 seems like the results are slightly harmed by more downsampling but curious if you basically get Proteina in the limit of doing no downsampling.
>
> We removed all the triangle layers and attention pair bias from both the AR and flow components, and also from the reproduced Proteina. This decision is made to pursure efficiency, and is supported by prior findings showing that models without triangle updates can still achieve strong designability, as demonstrated by Proteina. In our updated Table 1, our model attains **96.6% designability** without these components.
>
> In addition, removing these modules allows us to increase the batch size to 15 per GPU, avoiding compensating for a small batch size (4) by scaling to 128 GPUs as in Proteina. Finally, we agree with the reviewer's intuition that removing downsampling in PAR essentially recovers Proteina. In our preliminary experiments, both 1-scale PAR and Proteina produce consistent results across all evaluation metrics.
>
> >**`Q2:`** Does PAR hold to long length designability? Given its Proteina based the long length study could be extended here. It would be interesting to see if there is any value to being able to prompt a long length model.
>
> Thank you for this insightful suggestion. Long-protein generation is indeed a challenging frontier for protein design, and we have included this experiment below to provide more insights.
>
> **Finetuning on longer protein chains.** We follow Proteina to finetune our models on datasets with longer proteins. Since Proteina has not released its long-protein dataset and its statistics, we cannot entirely reproduce their experiment setups. Instead, we follow the filtering procedure described in their appendix on PDB structures to curate a long-protein dataset. We filter PDB structures to lengths between 256 and 768 residues and keep only designable samples, resulting in ~26k high-quality proteins. The length-distribution of this dataset (Fig. 9) exhibits a long-tail shape with peaks around 300-400 residues. After ligand removal, 6.7% of the samples fall below the 256-residue threshold. We then finetune the 400M PAR and Proteina models in Table 1 on this dataset for 10k steps.
>
> **Long-protein generation and oversampling.** We generate 100 proteins for each length in {300, 400, 500, 600, 700}. PAR exhibits higher designability at lengths {300, 400}, consistent with the higher density of training samples in this range. At lengths between 500 to 700, both Proteina and PAR show degraded designability, while PAR demonstrating slightly better results. We attribute this to the long-tail nature of the training set, which includes far fewer samples in the length range between 500 and 700. The limited size of the training set (26K) also potentially hinders the model's from reaching its full potential. We leave scaling up long-protein data as a promising direction for future work.
>
> These results are included in Table 8 and Figure 9 in the revised PDF.
>
> | length | 300 |  | 400 |  | 500 |  | 600 |  | 700 |  |
> | --- | --- | --- | --- | --- | --- | --- | --- | --- | --- | --- |
> |  | sc-RMSD | Designability (%) | sc-RMSD | Designability (%) | sc-RMSD | Designability (%) | sc-RMSD | Designability (%) | sc-RMSD | Designability (%) |
> | Proteina | 1.91 | 85 | 2.70 | 61 | 4.09 | 49 | 7.90 | 21 | 13.32 | 4 |
> | PAR | **1.28** | **93** | **1.65** | **72** | **3.19** | **52** | **6.80** | **29** | **11.29** | **10** |

---

> > ### Author Response · Authors · 2025-12-03
> >
> > > `W1:` No formal motif benchmark. Even if not SOTA it would be interesting to see.
> >
> > We quantify the **zero-shot** motif scaffolding performance of PAR along with other **training-based** approaches. The table below reports the ratio of unique solutions divided by total generated samples.
> >
> > We use PAR to generate 100 backbone structures for each benchmark problem in [1]. Following Proteina's evaluation protocol, we produce 8 ProteinMPNN sequences for each generation with the motif residues fixed, and feed each sequence to ESMFold. Using the predicted structure, we calculate ca-RMSD and MotifRMSD. A design is considered a success if any sequence achieves scRMSD ≤ 2Å, a motifRMSD ≤ 1Å, pLDDT ≥ 70, and pAE ≤ 5. All successful samples are clustered via hierarchical clustering and a TM-score threshold of 0.6 to obtain the number of unique solutions. For other training-based methods, we directly quote the results reported in Proteina, which evaluate results using 1000 samples.
> >
> > We have added the results in the Sec. C.8 of our paper.
> >
> > [1] RFDiffusion. Watson et al. Nature, 2023.
> >
> > |                                 | Zero-shot PAR (100 samples) | Proteina | Genie2 | RFDiffusion | FrameFlow |
> > |---------------------------------|:------------------:|:--------:|:------:|:-----------:|:---------:|
> > | 1PRW |          0         |     0.3    |    0.2   | 0.1 | 0.3 |
> > | 1BCF |          0         |     0.1    |    0.1   | 0.1 | 0.1 |
> > | 5TPN |          0         |     0.4    |    0.8   | 0.5      |  0.6     |
> > | 5IUS |          0         |     0.1    |    0.1   | 0.1      |     0     |
> > | 3IXT |          9.0         |     0.8    |   1.4   |      0.3      |     0.8     |
> > | 5YUI |          0         |     0.5    |    0.3   | 0.1  |     0.1     |
> > | 1QJG |          3.0         |     0.3    |    0.5   |      0.1      |     1.8    |
> > | 1YCR |          4.0         |    24.9   |   134  |      0.7      |    14.9    |
> > | 2KL8 |          4.0         |     0.1    |    0.1   |      0.1      |     0.1     |
> > | 7MRX.60 |       0      |     0.2    |    0.5   |      0.1      |     0.1     |
> > | 7MRX.85 |       1.0         |    3.1    |   2.3   |      1.3     |     2.2    |
> > | 7MRX.128 |     1.0         |    5.1    |   2.7   |      6.6     |     3.5    |
> > | 4JHW     |          0    |     0    |    0   |      0      |     0     |
> > | 4ZYP     |          0         |    1.1    |    0.3   |      0.6      |     0.4     |
> > | 5WN9     |          0         |     0.2    |    0.1   |      0      |     0.3     |
> > | 5TRV_short |      0        |     0.1    |    0.3   |      0.1      |     0.1     |
> > | 5TRV_med   |          0         |    2.2    |   2.3   |      1.0     |     2.1    |
> > | 5TRV_long  |          0         |    17.9   |   9.7   |      2.3     |     7.7    |
> > | 6E6R_short |          8.0         |    5.6    |   2.6   |      2.3     |     2.5    |
> > | 6E6R_med  |          2.0         |    41.7   |   27.2  |     15.1     |     9.9    |
> > | 6E6R_long |          3.0         |    71.3   |   41.5  |     38.1     |    11.0    |
> > | 6EXZ_short |          2.0         |     0.3    |    0.2   |      0.1      |     0.3 |
> > | 6EXZ_med  |          9.0         |    4.3    |   5.4   |      2.5     |    11.0    |
> > | 6EXZ_long   |         12.0         |    29.0   |   32.6  |     16.7     |    40.3    |
> > | # tasks with >= 1% success rate |         11         |    11    |   11   |      9      |     11    |

---

### Author Response · Authors · 2025-11-25
**General Response & Summary of Rebuttal**

Dear Reviewers, ACs and SACs,

We want to express sincere appreciation for all reviewers' efforts in reviewing and providing valuable suggestions! We have tried our best to address reviewers' concerns respectively, mainly including:

* Added analysis of inference speed and explored multi-scale orchestration of SDE/ODE for efficient sampling, as suggested by **all reviewers**.
* Added discussion about quantitative advantage over existing diffusion baselines, explored noise scaling parameter and finetuning-based approach for achieving state-of-the-art designability and distribution-level metrics, as suggested by Reviewer 8etu, SoNE, Zpd8.
* Added analysis of longer protein generation, as suggested by Reviewer Knjj and SoNE.
* Added motif scaffold benchmark results, as suggested by Reviewer Knjj and 8etu.
* Added clarifications on model architectures, as suggested by Reviewer Knjj.
* Added comprehensive ablation study of scale configurations, as suggested by Reviewer 8etu.
* Added clarifications on number of forward passes and the process of scheduled sampling, as suggested by Reviewer 8etu.
* Added ablation study of AR encoder, decoder size, and discussed the limitations of per-token decoder, as suggested by Reviewer SoNE.
* Added analysis on whether the proposed 1D sequence-based downsampling preserves spatial relationships, as suggested by Reviewer SoNE.
* Added discussion about our rationale of data space modeling over VAE/VQ-VAE based approaches, as suggested by Reviewer SoNE.
* Added discussion about challenges and novelty of applying AR models in protein structure generation, as suggested by Reviewer SoNE and Zpd8.
* Added essential technical details about multi-scale downsampling/upsampling, motif scaffold superimposing, and scale-level attention map calculations, as suggested by Reviewer Zpd8.
* Added analysis on scale-agnostic inference and discussed the robustness of PAR when inferring with a different scale configuration, as suggested by Reviewer Zpd8.

We again thank everyone's time and effort in discussing, providing valuable and inspiring feedback, and helping us improve our manuscript. Moreover, we have accordingly revised the paper to best include most of the insightful suggestions and comments from the reviewers. We do sincerely appreciate you!

We are very happy to address any further feedback during the discussion phase!

Many thanks, and cheers!

Authors

---

### Author Response · Authors · 2025-12-03
**Summary of Rebuttal to Support Evaluation**

Dear ACs and SACs,

We would like to best support your evaluation process in light of the challenges ICLR faced this year. Below we summarize our contributions and responses to reviewers’ concerns to facilitate assessment. We sincerely appreciate your valuable time and suggestions.

**Contributions**
1. We present PAR, the *first* multi-scale AR model for protein backbone generation, addressing limitations of existing AR methods.
2. PAR comprises multi-scale downsampling, AR transformer, and a flow-based decoder to model Cα atoms directly, avoiding discretization loss.
3. Exposure bias is mitigated via *noisy context learning* and *scheduled sampling*, improving structure generation.
4. PAR shows an interpretable generation process that forms coarse backbone topology followed by refinement.
5. Strong generation quality: FPSD 161.0, designability 96.6%.
6. Demonstrates zero-shot generalization, showing versatility for conditional tasks like prompt-based generation.


**Strengths Suggested by Reviewers**
| Reviewer | Strength |
| --- | --- |
| **Knjj** | **High technical novelty**; competitive performance; clear, well-written; AR process done without tokenization is *elegant*. |
| **8etu** | First multi-scale AR model for protein backbone generation; noisy context learning + scheduled sampling mitigate exposure bias; demonstrates strong **zero-shot generalization**. |
| **SoNE** | Highly innovative: multi-scale AR + flow decoder in continuous space; demonstrated flexible user-guided generation. |
| **Zpd8** | High-quality, well-written, and clear paper; coarse-to-fine sculpting is interesting and offers a new AR path in this domain. |



\
**Addressed Concerns**

We appreciate all the constructive feedbacks and have tried our best to address reviewers' concerns respectively.

| Question | Knjj | 8etu | SoNE | Zpd8 | Action & Summary |
| --- | --- | --- | --- | --- | --- |
| Inference speed | ✔️ | ✔️ | ✔️ | N/A | Introduced runtime analysis and multi-scale SDE/ODE, achieving **1.96x–2.5x** sampling speedup compared to the single-scale baseline. |
| Performance vs baselines | N/A | ✔️ | ✔️ | ✔️ | Clarified superior FPSD, explored noise scaling & finetuning, achieving **SOTA** in FPSD & designability. |
| Long protein generation | ✔️ | N/A | ✔️ | N/A | Added experiments; PAR outperforms baseline; discussed future improvements for >500 length. |
| Motif scaffold benchmarking | ✔️ | ✔️ | N/A | N/A | Benchmarked **zero-shot** motif scaffolding. |
| Architecture clarifications | ✔️ | N/A | N/A | N/A | Clarified our removal of triangle layers & pair bias; improved efficiency and batch size. |
| Scale ablation | N/A | ✔️ | N/A | N/A | Tested 2–5 scales; 3-scale is optimal. |
| Forward passes & scheduled sampling | N/A | ✔️ | N/A | N/A | Detailed scheduled sampling process and number of forward passes per scale. |
| Encoder/decoder size | N/A | N/A | ✔️ | ✔️ | Ablated AR encoder size; discussed challenges of per-token decoder and potential lightweight decoder design. |
| Sequence downsampling & spatial correlations | N/A | N/A | ✔️ | N/A | Sequence downsampling preserves essential pairwise correlations; Analyzed lddt & RMSE of pairwise distance maps. |
| Data vs latent-space | N/A | N/A | ✔️ | N/A | VQ-VAE suffers fidelity loss; Discussed VAE-based and hybrid approaches. |
| Technical clarifications | N/A | N/A | N/A | ✔️ | Clarified down/up-sampling, motif superimposing, scale-level attention maps. |
| Scale-agnostic inference | N/A | N/A | N/A | ✔️ | Scale-agnostic inference works with different scale configuration beyond training via removing scale-specific embedding. |

\
We again thank your time and effort in providing valuable and constructive feedback when evaluating our manuscript. We have accordingly revised the paper to include most of the insightful comments from the reviewers. We sincerely appreciate you once again.

Many thanks,
Authors

---

### Meta-Review · Area_Chair_3yG9 · 2026-01-07

**Summary:**

* Lack of motif scaffolding benchmark
* No ablation over number of hierarchical scales
* Only demonstrated on short proteins <256 AA
* Inference speed comparison
* No demonstrated advantages over existing diffusion-based protein generative models in terms of quality or efficiency
* Leans heavily on pre-existing flow-based model making it difficult to assess the true contribution of the AR components, feels incremental over existing methods.

In general, the reviewers were concerned that the additional complexity involved was not reflected in improved performance in terms of either quality or efficiency.

**Reviewer Concerns:**

All asked for experiments were added in the rebuttal. The authors made efforts to further tune PAR to improve performance especially for table 1.

I have two concerns remaining about the additional experiments.

* **Improvements of PAR** using parameter tuning of $\gamma$ and finetuning on DB. While designability is improved, diversity is worse. This is a bit concerning because all models are able to trade off diversity for designability. The original results were presented at approximately equivalent levels of diversity. These "improved" versions of PAR have fairly significantly worse diversity. The updated paper claims that PAR now outperforms "all diffusion-based baselines". I'm not sure this claim is fully supported.

* **Ablations on Longer proteins + Sampling speed** in the rebuttal, experiments on larger proteins and with different numbers of steps performance is measured in terms of Designability and sc-RMSD. This metrics alone cannot tell the full story of performance, as its very easy for these types of models to mode collapse (especially with fewer steps) and achieve better designability be producing the same protein over and over.

**Reviewer Scores:**

In my view, reviewer Knjj would have increased their score as both their concerns and questions were answered.

The other reviewers would likely not have changed their scores. Even after the updated, it is not clear there is any significant improvement over the base Proteina model in terms of performance or efficiency. More data is needed than is supplied in the rebuttal to conclude this.

It is possible after extended discussion that these issues could have been resolved. It is unfortunate that this was unable to happen in this conference.

---

### Decision · Program_Chairs · 2026-01-26

Reject